# TGF-β inhibitor treatment of H$_2$O$_2$-induced cystitis models provides biochemical mechanism for elucidating interstitial cystitis/painful bladder syndrome patients

Hideto Taga[1,2], Tsunao Kishida[2], Yuta Inoue[1,2]*, Kenta Yamamoto[2], Shin-ichiro Kotani[2], Tsujimoto Masashi[1,2], Osamu Ukimura[1], Osam Mazda[2]

1 Department of Immunology, Kyoto Prefectural University of Medicine, Kamigyo-ku, Kyoto, Japan,
2 Department of Urology, Kyoto Prefectural University of Medicine, Kamigyo-ku, Kyoto, Japan

* u-turn-n@koto.kpu-m.ac.jp

## Abstract

Interstitial cystitis/painful bladder syndrome (IC/PBS) is a chronic disease for which no effective treatment is available. Transforming growth factor-β (TGF-β) is thought to be involved in the pathogenesis of IC/PBS, and previous studies have suggested that administrations of a TGF-β inhibitor significantly ameliorated IC/PBS in a mouse model. However, the molecular mechanisms underlying the therapeutic effect of a TGF-b inhibitor on IC/PBS has not been comprehensively analyzed. TGF-β has a variety of actions, such as regulation of immune cells and fibrosis. In our study, we induced IC/PBS-like disease in mice by an intravesical administration of hydrogen peroxide (H$_2$O$_2$) and examined the effects of three TGF-β inhibitors, Repsox, SB431542, and SB505124, on the urinary functions as well as histological and gene expression profiles in the bladder. TGF-β inhibitor treatment improved urinary function and histological changes in the IC/PBS mouse model, and SB431542 was most effective among the TGF-β inhibitors. In our present study, TGF-β inhibitor treatment improved abnormal enhancement of nociceptive mechanisms, immunity and inflammation, fibrosis, and dysfunction of bladder urothelium. These results show that multiple mechanisms are involved in the improvement of urinary function by TGF-β inhibitor.

## Introduction

Interstitial cystitis/painful bladder syndrome (IC/PBS) is a chronic disease characterized by bladder pain, frequent and urgent urination, and nocturia. The known prevalence of IC/PBS or conditions suggestive of IC/PBS ranged from 0.01% to 2.3%, of the population, with an approximately five-fold dominance among the women. IC/PBS is an intractable illness for which there is currently no effective treatment [1].

IC/PBS is considered to be caused by various pathological mechanisms including dysfunction of urothelium, activation of lymphocytes and mast cells, bladder inflammation, abnormal enhancement of nociceptive mechanisms, toxic urine substances, microbial infection, etc. IC/

**Data Availability Statement:** All RNAsequence files are available in National Center for Biotechnology Information (NCBI) Gene

Expression Omnibus (GEO) repository, [Accession Number: GSE226802].

**Funding:** The authors received no specific funding for this work.

**Competing interests:** The authors have declared that no competing interests exist.

PBS is divided into three groups: Hunner type interstitial cystitis (HIC), Non-Hunner type interstitial cystitis (NHIC), and hypersensitive bladder (HSB). HIC has been reported to be accompanied by significant lymphocytic infiltration and urothelial cell denudation in the bladder, which are not seen in NHIC and HSB [1].

A commonly used animal model for IC/PBS is cyclophosphamide (CP)-induced cystitis in mice [2]. An intraperitoneal injection of CP induces acute hemorrhagic cystitis with bladder overactivity. However, this cystitis model is characterized by acute inflammation rather than chronic pathology. In the CP-induced cystitis, inflammation sustains for only short periods and ceases within several days. Recently, a mouse model using $H_2O_2$ has been reported. Even a week after an infusion of $H_2O_2$ into the bladder, urination frequency remains high, and the high frequency of urination lasts for longer periods than that in CP model [3].

Transforming growth factor-β (TGF-β) has been reported to increase in the bladder tissue of IC/PBS patients [4]. Some previously studies showed that an intravesical administration of TGF-β receptor inhibitor (SB505124) 48 hours after induction of cystitis decreased the frequency of urination and increased one-time voiding volume in mice. They also reported that TGF-β inhibitors reduced ATP release from urothelial cells and inhibited bladder afferent nerve discharge via stimulation of purine receptors [5,6]. However, TGF-β signaling activates multiple and complicated intracellular pathways and affect the expression regulation of a variety of genes [7,8]. The overall mechanism is not clear as to how the inhibition of TGF-β signaling ameliorates the cystitis. In addition, long-term effect of TGF-β inhibitors remains to be clarified. Therefore, we administered various TGF-β inhibitors to $H_2O_2$-induced cystitis mice, and analyzed urinary disturbance for 7 days. We also performed comprehensive transcriptome analyses to elucidate the overall mechanisms of the treatment of $H_2O_2$-induced cystitis by TGF-β inhibitors.

## Materials and methods

### Chemical compounds

SB431542 and SB505124 were purchased from Cayman chemical company (Ann Arbor, MI, USA). Repsox was purchased from focus biomolecules (Plymouth Meeting, PA, USA).

### Cystitis mouse model and treatment with TGF-β inhibitor

All animal experiments were approved by the Committee for Animal Research, Kyoto Prefectural University of Medicine (M2021-329), and the care of the animals was performed according to the institutional guideline and Guide for the Care and Use of Laboratory Animals. Female C57BL/6N mice were purchased from SHIMIZU Laboratory Supplies (Kyoto, Japan). They were housed in groups of 4 per cage in a room maintained at 23˚C and 50% relative humidity with an alternating 12-h light/dark cycle (the lights came on automatically at 7:00 a. m.). Food and water were freely given. The $H_2O_2$-induced cystitis model was prepared as previously reported, with slight modification [3]. Briefly, 30% $H_2O_2$ (Nacarai tesque) was diluted with sterile saline to 1.5% immediately before use. The mice at the age of 8 weeks were anesthetized with isoflurane. A 24-gauge catheter was inserted into the bladder through the urethra and 100μL of 1.5% $H_2O_2$ solution was infused into the bladder for 30 minutes. Urinary frequency and one-time voiding volume were measured on days 1 and 8 to evaluate the effect of the $H_2O_2$ irritation on urinary function. Thirty min after $H_2O_2$ emission, 1.2 mM TGF-β inhibitors (either SB431542, SB505124, or Repsox) were intraperitoneally injected at 100μl/mouse. TGFβ inhibitor concentrations were determined based on previous literature [9–11]. In this experiment of TGF-β inhibitor treatment, 100μL of saline was infused into the bladder of the control mice. On day 4, an injection of TGF-β inhibitors were repeated.

## Mouse voiding behavior analysis

Urination was recorded by the Voided Stain On Paper (VSOP) method as described with slight modifications(S1 Fig) [12]. To avoid micturition variation resulting from circadian rhythms, functional tests were conducted at similar times of the day (between 9:00 AM and 1:00 PM). Briefly, each mouse was placed in a certain area above a filter paper (Advantec 300 mm; Tokyo Roshi Kaisha, Ltd., Tokyo, Japan). The voiding time and area were recorded for 3 hours. The recorded urine stains were scanned into image files and the stained areas were calculated using the software ImageJ v1.47 (National Institutes of Health, Bethesda, MA, USA).

## Hematoxylin and eosin (HE) staining

Mice were sacrificed by intraperitoneal administration of 100–150 mg/kg of pentobarbital and bladder was excised from the mice 3 hours, 1 day and 8 days after $H_2O_2$ infusion. Bladder tissue was dissected in an axial direction. The tissue was fixed in 4% paraformaldehyde (PFA). After fixation in 4% PFA some parts of the organ were cryoprotected in 30% sucrose PBS (-) and freeze-mounted. The specimens were sliced into 10μm thick sections and stained with hematoxylin and eosin.

## RNA-seq analyses

Total RNA was extracted from a hemisphere of the bladder using the ISOGEN II (NIPPON GENE CO., LTD) according to the manufacturer's instructions. mRNA was purified from total RNA using poly-T oligo-attached magnetic beads, and after random fragmentation, cDNA was synthesized. The qualified libraries were sequenced with Illumina Novaseq 6000 (Illumina; paired-end, 150 bp). Analysis was performed using strand NGS (https://www.strand-ngs.com/).

## Reverse transcription-quantitative PCR (RT-qPCR)

Total RNA was extracted from the bladder as above. cDNA was synthesized using ReverTra Ace qPCR (TOYOBO Life Science), and subjected to real-time RT-PCR using StepOnePlus Real-Time PCR Systems (Applied Biosystems). The reaction mixture included the Taqman probe (Applied Biosystems), Taqman Fast Advanced Master Mix (Applied Biosystems) and probe/primers as described below. The relative mRNA levels were calculated as follows: Relative mRNA level (fold) = ((target gene mRNA level in sample) / (*β-actin* gene mRNA level in sample)) / ((target gene mRNA level in control) / (*β-actin* gene mRNA level in control)). Probes for *TGF-β1* and *TGF-β2* genes were purchased from Roche (Universal Probe Library). Sequences of *TGF-β1* and *TGF-β2* gene primers were as follow: forward primer for *TGF-β1* gene: 5´-TGGAGCAACATGTGGAACTC-3´; reverse primer for *TGF-β1* gene: 5´-GTCAGCAGCCGGTTACCA-3´; forward primer for *TGF-β2* gene: 5´-GAGAAAAATGCTTCGAATCTGG-3´, reverse primer for *TGF-β2* gene: 5´-TGGGAGATGTTAAGTCTTTGGAT-3´; forward primer for *β-actin* gene: 5´-CTAAGGCCAACCGTGAAAAG-3´, reverse primer for *β-actin* gene: 5´-ACCAGAGGCATACAGGGACA-3´. Primers and probes for *Trpa1*, *IL-1β*, *Col7a1*, *Krt13*, and *Krt6a* genes were purchased from Applied Biosystems.

## Azan staining

Bladder was excised from the mice 10 days after $H_2O_2$ administration. Bladder was cut at the maximum plane in the axial direction and tissue sections were subjected to Azan staining.

The specimens were sliced into 10μm thick sections. Firstly, the tissues were mordanted for 30 min in the mordant solution which consists of 10% potassium dichromate and 10% trichloroacetic acid. Next, they were washed with water, the tissues were incubated in amocarzine G solution at room temperature for 30 min, washed with water, immersed in 1% aniline ethanol, washed with acetate alcohol, washed with water, and they were stained in 5% phosphotungstic acid overnight. Next day, they were washed with water and stained in aniline blue-orange G solution for 1 hour. Subsequently, they were dehydrated in absolute ethanol, transparentized in Xylene, mounted with neutral resin. The percentage of bladder tissue affected by fibrosis was determined by calculating the ratio of collagen area to smooth muscle area. Quantitative analysis of the collagen area was performed using an image analyzer system (Fiji 2.5.0; National Institutes of Health, Bethesda, MA, USA).

## Statistical analyses

Data are expressed as means +/− SD. Statistical significance was analyzed by one-way ANOVA with the Tukey-Kramer post hoc test. $P < 0.05$ was considered significant. All analyses were conducted with EZR.

## Results

### TGF-β inhibitors improved the urinary function and morphological changes in the $H_2O_2$-induced cystitis model

As shown in Fig 1A, $H_2O_2$-treated mice showed a significant increase in urinary frequency and a significant decrease in one-time voiding volume compared with non-treated and saline-treated mice. In $H_2O_2$-treated mice, the bladder urothelium was thin and damaged, and edematous thickening was observed in the submucosa 3 hours and 1 day after the $H_2O_2$ infusion, compared with saline-treated mice (Fig 1B). The urinary dysfunction was recovered to almost normal by the 8th day after the $H_2O_2$ infusion, while edematous thickening of the submucosal layer was seen even on day 8 (Fig 1B). In previous report, the significant elevation of *IL-1β* mRNA expression was observed on the 7th day after the $H_2O_2$ infusion [3]. In the $H_2O_2$-induced cystitis model, the significant elevation of *IL-1β* mRNA expression was observed on day 14 (S2 Fig). Also, to use TGF-β inhibitors for the $H_2O_2$-induced cystitis model, *TGF-β1* and *TGF-β2* mRNA expression were measured. *TGF-β1* and *TGF-β2* mRNA expression was significantly elevated day 2 after the $H_2O_2$ infusion compared to saline-infused mice. *TGF-β1* mRNA significantly decreased on day 4 compared to day 2 mice (S2 Fig).

Next, we examined the effects of TGF-β inhibitors on $H_2O_2$-induced cystitis in mice. Three TGF-β inhibitors, Repsox, SB431542, and SB505124, or saline was injected to the cystitis mice 30 min and 4 days after the $H_2O_2$ infusion as shown in Fig 2. As shown in Figs 3A and S3, the intraperitoneal administration of all three TGF-β inhibitors significantly improved the tidal voiding volume on day 7 (Fig 3A). Repsox and SB431542, but not SB505124, significantly reduced the urinary frequency (Fig 3B). As shown in Fig 3C, any significant difference was not seen among the total urine volumes of all the groups, suggesting that the increase in urinary frequency of the TGF-β-treated mice was not due to high total urine volume. These results strongly suggest SB431542 as the most effective TGF-β inhibitor for the treatment of impaired urinary functions in the cystitis mice.

### Possible involvement of inflammatory and neural signaling-related genes in the therapeutic effect of SB431542 on mouse cystitis

To find out the molecular mechanisms through which a TGF-β inhibitor alleviated the cystitis, we performed RNA sequencing (RNA-seq) analysis for the bladder tissue of control (sham),

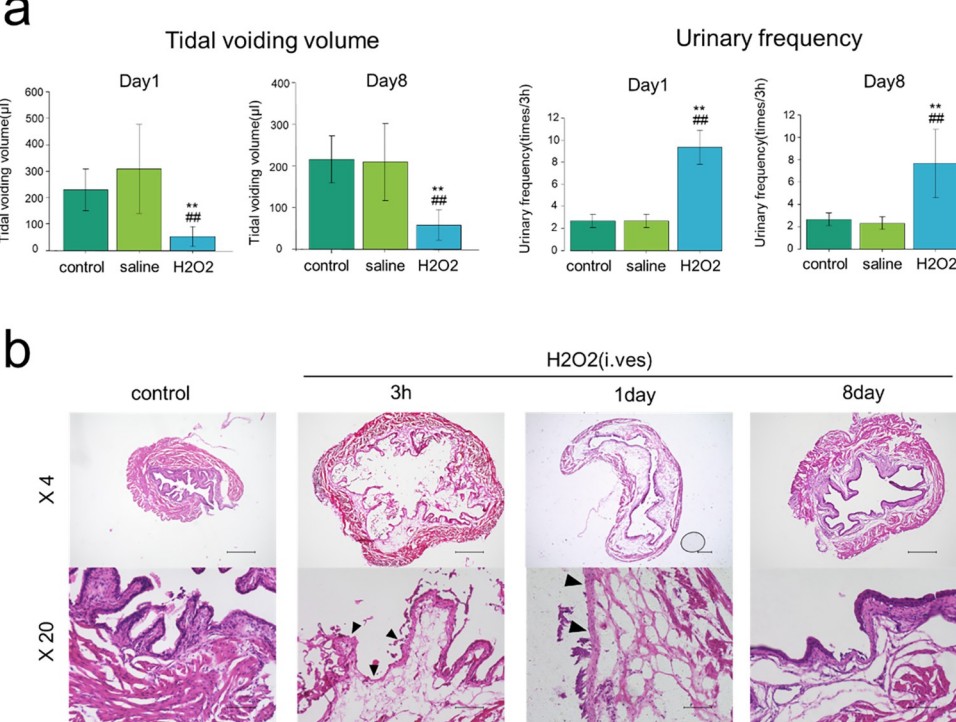

**Fig 1. Intravesical infusion of H₂O₂ caused IC/PBS-like urinary disturbance and histological changes in mice.**
Mice were given an intravesical infusion of 1.5% H₂O₂ or saline through the urethra as described in the Materials and Methods. Control mice were inserted with a catheter through the urethra. (a) One and 8 days later, mice were subjected to voiding behavior analysis. Means ± sd. of tidal voiding volume (left) and urinary frequency (right) are shown (N = 3 mice for each group). **P<0.01 vs. Control. ##P<0.01 vs. Saline. (b) After indicated periods, bladder was excised and tissue sections were stained with HE. Representative images of low (top panels) and high (bottom panels) magnifications are shown. Scale bars show 500 μm (top panels) and 100 μm (bottom panels).

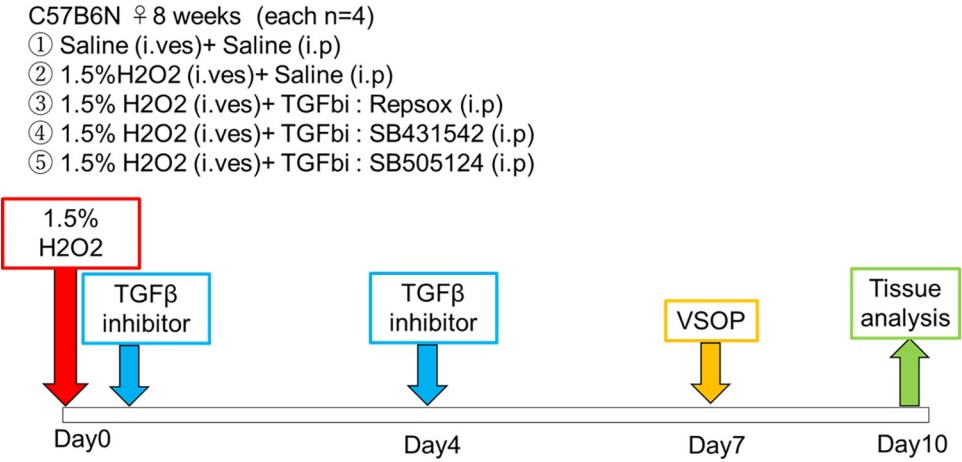

**Fig 2. Experimental scheme is shown.** The mice at the age of 8 weeks were anesthetized with isoflurane. 1.5% H₂O₂ were intravesically infused into the bladder for 30 minutes. Thirty min after H₂O₂ emission, TGF-β inhibitors (either SB431542, SB505124, or Repsox at 1.2 mM, 100μl/mouse) were intraperitoneally injected. On day 4, an injection of TGF-β inhibitors were repeated. Urination was recorded on day 7 by VSOP. Mice were sacrificed and bladder was removed on day 10.

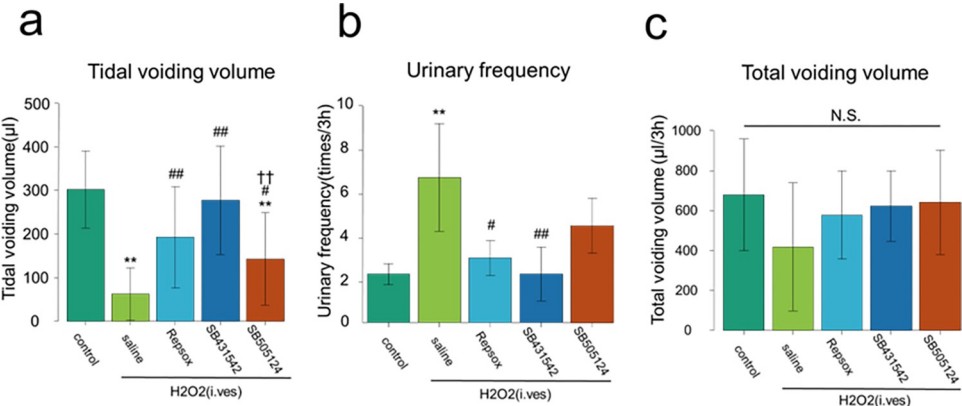

**Fig 3. TGF-β inhibitor treatment alleviated urinary disturbance in cystitis mice.** Mice were given an intravesical infusion of 1.5% $H_2O_2$ solution on day 0. On days 0 and 4, TGF-β inhibitors or saline were injected into the peritoneal cavity of mice. On day 7, voiding behavior analysis was performed as described in the Materials and Methods. Means ± sd. of tidal voiding volume (a), urinary frequency (b) and total voiding volume (c) are shown (N = 3 mice for each group). **$P<0.01$ vs. Control. #$P<0.05$, ##$P<0.01$ vs. Saline ††$P<0.01$ vs. SB431542. N.S.: Not significant among the groups.

$H_2O_2$/saline-treated, and $H_2O_2$/SB431542-treated mice. Principal component analysis (PCA) revealed that the three groups were separately located with sufficient variances (X, Y and Z axes: 35.67%, 29.43% and 16.45% contribution, respectively), strongly suggesting that the $H_2O_2$ stimulus caused considerable aberration in gene expression profiles in the bladder (Fig 4A). The dots of the $H_2O_2$/SB431542-treated group were located midway between the control and the $H_2O_2$/saline-treated groups, indicating that the administration of SB431542 remarkably restored the aberrant mRNA expression in the inflamed bladder. Fig 4B shows scatter plots. The correlation coefficient between the control and the $H_2O_2$/SB431542-treated groups was the highest (R2 = 0.932), followed by that between the $H_2O_2$/SB431542-treated and the $H_2O_2$/saline-treated groups (R2 = 0.878). The comparison between the control and the $H_2O_2$/saline-treated groups represented the lowest correlation coefficient value (R2 = 0.765). Thus, it was considered that the irritable effects of $H_2O_2$ were counteracted by the SB431542 administration through the normalization of the aberrant gene expression. Among the Gene Ontology (GO) terms categorized into "biological process", we listed ten GO terms that represent gene sets differentially expressed between two groups (Fig 4C). Nine of the ten GO terms for the gene sets that showed highest p values for control group vs. $H_2O_2$/saline-treated were related to "immunity and inflammation", suggesting that the effects of $H_2O_2$ were heavily associated with "immunity and inflammation". In addition, four of the ten GO terms for the gene sets showing highest p values for $H_2O_2$/saline vs. $H_2O_2$/SB431542-treated groups were related to in "synaptic signal", which may imply that synaptic signals may play important roles in the effects of SB431542 on the cystitis. There was also a significant difference in expression levels of the genes involved in the "immunity and inflammation" between the control and $H_2O_2$/SB431542-treated groups, but the p-value was not as low as that for the difference for the control vs. $H_2O_2$/saline-treated groups. Based on the results of GO analyses, we performed heatmap and hierarchical clustering analyses for genes involved in "bladder afferents nerve" and those involved in "immunity and inflammation", as well as for genes representing "keratin" and "collagen" that may be involved in the pathogenesis of IC/BPS and the effect of TGF-β inhibitors (Fig 4D). The heatmap and hierarchical clustering analysis revealed that the $H_2O_2$/SB431542-treated group is more closely similar to the control group than the $H_2O_2$/saline-treated group with respect to the genes related to "bladder afferents nerve" and "immunity and

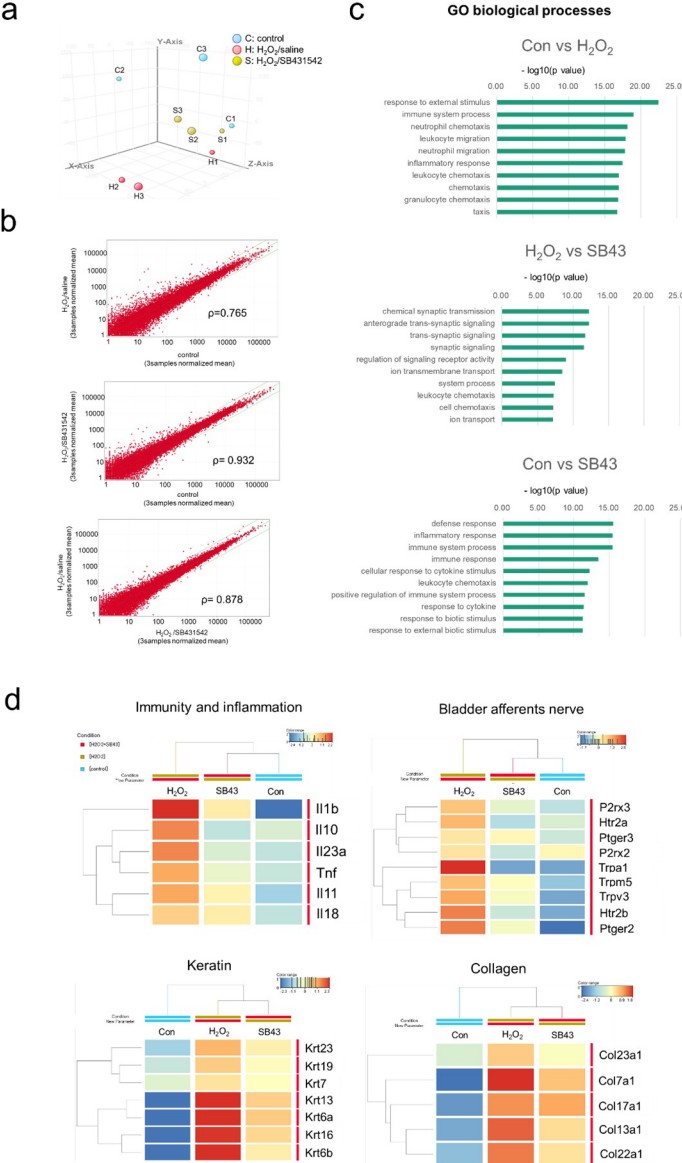

**Fig 4. RNA-seq analyses suggest molecular mechanisms through which SB431542 alleviated cystitis.** $H_2O_2$-treated mice received intraperitoneal injections of SB431542 or saline as in Fig 3. On day 10, RNA was extracted from the bladder and examined by RNA sequencing. (a) Data of principal component analysis are shown. Blue, red and yellow dots represent control, $H_2O_2$/saline-treated, and $H_2O_2$/SB431542-treated groups, respectively. (b) Scatter plots are shown. Green lines show changes in expression with p = 0.01.ρ: Spearman's rank correlation coefficient (control group vs. $H_2O_2$/SB431542-treated group: R2 = 0.932, $H_2O_2$/saline-treated group vs. $H_2O_2$/SB431542-treated group: R2 = 0.878, control $H_2O_2$/saline-treated group vs. $H_2O_2$/saline-treated group: R2 = 0.765). (c) Biological processes suggested by Gene Ontology (GO) analysis are shown. GO terms that were changed more than 2-fold were selected, and the top 10 GO terms with the lowest p-values are drawn. (d) Heat maps and hierarchical clustering analyses comparing control, $H_2O_2$/saline and $H_2O_2$/SB431542 groups are shown.

inflammation". The gene sets related to "keratin" and "collagen" were highly expressed in the $H_2O_2$/saline-treated group, but their expression was decreased by SB431542 administration, as demonstrated by the similarity between the heat maps for the control and $H_2O_2$/ SB432542-treated groups.

## $H_2O_2$ infusion augmented expression of *Trpa1*, *IL-1β*, *Col7a1*, *Krt13*, and *Krt6a* genes in the bladder, which was reversed by SB431542 administration

To confirm the results from the heatmap and the hierarchical clustering analyses, we performed reverse transcription-quantitative PCR (RT-qPCR) and validated expression levels of the representative genes that were categorized in the "bladder afferents nerve", "immunity and inflammation", "keratin", and "collagen". We found that mRNA for the *Trpa1* gene that is involved in bladder afferents was highly expressed in the bladder of the $H_2O_2$/saline-treated mice, whereas the *Trpa1* mRNA was significantly less abundant in the bladder of control and the $H_2O_2$/SB431542-treated mice. Similar results were also obtained by evaluating mRNA for the inflammatory cytokine *IL-1β*, *Krt13* and *Krt6a* that are expressed in urothelial cells, and *Col7a1* that is involved in fibrosis (Fig 5).

## SB431542 administration ameliorated bladder fibrosis in the $H_2O_2$-induced cystitis model

To determine the degree of fibrosis, the bladder tissue sections were subjected to AZAN staining that enables us to distinguish between collagen fibers (blue) and muscles (red). The proportion of the fibrosis area in each bladder tissue specimen was calculated (Fig 6). The $H_2O_2$/saline-treated group higher proportion of fibrosis area in the bladder than the control group, while the proportion in the $H_2O_2$/SB431542-treated group was comparable to that in the control group.

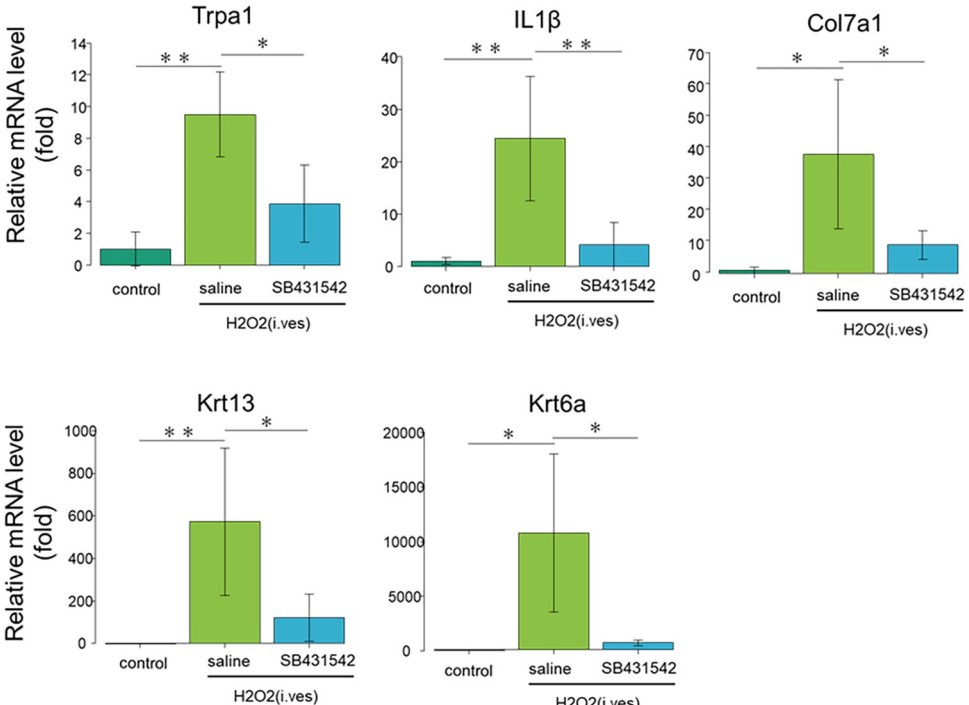

**Fig 5. SB431542 treatment restored enhanced expression of several genes.** Bladder was excised from control, $H_2O_2$/saline-treated and $H_2O_2$/SB431542-treated mice as in Fig 4. RNA was extracted from the bladder, and RT-qPCR validation of the indicated genes were performed. Shown are as means ± sd. of relative mRNA levels (n = 4 mice). *p<0.05 and **p<0.01 between the indicated groups.

**Fig 6. SB431542 treatment inhibited fibrosis of the bladder in H₂O₂-induced cystitis mice.** Bladder tissue sections were stained with Azan, and microscopic images were analyzed as described in the Materials and Methods. (Left), microscopic images are shown (control group vs. H₂O₂/saline-treated group, control group vs. H₂O₂/SB431542-treated group, H₂O₂/saline-treated group vs. H₂O₂/SB431542-treated group; p = 0.0000764, p = 0.8656326, p = 0.0001310, respectively). Scale bar = 1,000μm. (Right), Means ± sd. of ratios of (collagen-stained area)/(total area) were calculated (n = 4 mice per group). **p<0.01 between the indicated groups.

## Discussion

Systemic administration of a TGF-β inhibitor improved disturbance of urinary function in a mouse model of IC/PBS. The present results can contribute to the elucidation of molecular pathophysiology of IC/PBS and mechanisms of therapeutic effect of TGF-β inhibitors on IC/PBS [7].

Our RNA-seq analyses results suggest that some genes categorized into four gene sets, i.e., "bladder afferents nerve", "immunity and inflammation", "keratin", and "collagen" may have contributed to the therapeutic effect of SB431542 on the cystitis.

One of the mechanisms of bladder pain induction in IC/PBS is assumed to be chronic tissue inflammation that causes functional changes in C-fiber afferents [13]. In particular, *Trpa1*, a polymodal receptor, has been identified not only in C-fiber afferents, but also in urothelial cells and detrusor muscle cells in the urinary bladder. It has been confirmed that an administration of a TRPA1 channel activator into the bladder induces detrusor overactivity [14].

The H₂O₂ stimulus causes oxidative stress in the bladder, leading to activation of the afferent pathway of capsaicin-sensitive C-fibers, which subsequently results in hyperactivity of the detrusor muscles [15,16]. Oyama et al. reported that mRNA expression of *Trpa1* in bladders of the H₂O₂-treated mice was significantly higher than that in the control mice [17]. In the present study, the overexpression of the genes that could be involved in afferent nerve of the bladder was restored by SB431542 treatment, as shown by the heatmap of RNA-seq analyses (Fig 4D). RT-qPCR indicated that the mRNA level for *Trpa1* was elevated in the cystitis and was decreased by SB431542 treatment (Fig 5).

Recent RNA-seq analyses suggested that immunologic inflammatory processes might support the pathophysiology of Hunner type interstitial cystitis [18]. TGF-β has opposing pro- and anti-inflammatory effects. TGF-β knockout mice displayed lethal inflammation, whereas an immunosuppressive function of TGF-β is also reported [19,20]. Meanwhile TGF-β may conversely play pro-inflammatory effects on Th17 cells and cells of the innate immune system. TGF-β, together with interleukin-6 (*IL-6*), was reported to be an essential player in driving pro-inflammatory Th17 lineage differentiation [21–23]. In addition, patients with autoimmune diseases often develop IC/BPS as a complication [24]. Th17 cells, which is causally associated with Hunner type interstitial cystitis, have also been implicated in autoimmune diseases such as rheumatoid arthritis, multiple sclerosis, psoriasis, and inflammatory bowel disease [25–27]. A previous report suggested that inhibition of the TGF-β signaling pathway may be beneficial in patients with autoimmune disorders, such as multiple sclerosis, through downregulation of the Th17 pathway [28]. Minami et al. reported that mRNA expression of *IL-1β* in

bladders of the $H_2O_2$-treated mice was significantly higher than that in the control mice [29]. In the present study, the heatmap and hierarchical clustering analysis showed that TGF-β inhibitor treatment partially restored the expression of the genes that are involved in immunity and inflammation, as shown in the Fig 4D. Also, the mRNA level for *IL-1β* was elevated in the cystitis and was decreased by SB431542 treatment (Fig 5). However, GO analysis indicated that there was still an inflammatory difference between control and $H_2O_2$/SB431542-treated group. In other words, it can be concluded that the administration of TGF-β inhibitors partially, but not completely, suppressed inflammation.

TGF-β inhibitors have been reported to inhibit fibrosis in various diseases [7]. There was a report that a TGF-β receptor inhibitor was administered to rats with ketamine-induced bladder fibrosis and it inhibited the bladder fibrosis of the mouse [30]. Minami et al. showed that fibrosis area in bladder submucosa of the $H_2O_2$-treated mice increased than that of the control mice [29]. In IC/BPS patients, collagen staining area of the bladder is increased. The bladder fibrosis is associated with decreased bladder compliance, uncontrolled detrusor contractions, increased frequency of urination, and decreased bladder capacity [31–34]. In the present study, TGF-β inhibitor administration improved the expression of genes related to fibrosis as shown in the heatmap of RNA-seq analyses. Moreover, SB431542 treatment inhibited fibrosis of the bladder in cystitis (Fig 6). RT-qPCR indicated that the mRNA level for *Col7a1*, of which mutation is known to cause hereditary epidermolysis bullosa, was elevated in the cystitis and was decreased by SB431542 treatment [35]. Although *Col7a1* has not been reported as a mediator of $H_2O_2$-induced cystitis, the results of increased *Col7a1* expression in $H_2O_2$-induced cystitis mice are consistent because the bladder wall of $H_2O_2$-induced cystitis mice has been shown to have increased fibrosis [29].

Urothelial denudation is another characteristic histological feature of Hunner type interstitial cystitis. The loss of the urothelial barrier could permit the stimulation of urine to directly contact the afferent nerves in the bladder [18]. Since a defect in the urothelium is thought to be one of the causes in IC/PBS, there is a report of transplantation of urothelial cells created by direct conversion into $H_2O_2$ models [36]. Keratin is one of the cytoskeletons expressed in epithelial cells, including the urothelium of the bladder. Laguna et al. reported that keratin profiles of IC/BPS and normal bladder are different from each other [37]. *Krt13* is a keratin that can be expressed in normal urothelium and in IC/BPS to the same extent, but *Krt6a*, a component of basal keratin, is expressed in urothelium in IC/PBS but not in normal urothelium [37–39]. In the present study, mRNA levels for *Krt13* and *Krt6a* were increased by $H_2O_2$ administration and decreased by TGF-β inhibitor administration, suggesting that TGF-β inhibitor may have protected urothelial cell damage by $H_2O_2$ stimulus. *Krt6* and *Krt13* has not been reported as a mediator of $H_2O_2$-induced cystitis. The *Krt6* expression is seen in stratified epithelial cells featuring hyperproliferation, such as psoriasis. *Krt13* is associated with epithelial cell differentiation and signal transduction [40]. $H_2O_2$ destroys urothelial cells, followed by regeneration of urothelial cells through a process of proliferation and differentiation. Therefore, it seems that these increases in mRNA expression of keratins in $H_2O_2$-induced cystitis mice are consistent. According to previous reports, proteins involved in urothelial barrier function include uroplakins, proteoglycan core proteins, tight junction proteins, and claudins [1,41]. Uroplakin knockout mice have been used as mouse models of IC/PBS [41]. S4 Fig shows that the mean raw reads count of $H_2O_2$-treated mice was lower than that of control mice and SB43 mice, except for *Upk1b*. The mean raw reads count of *UPK3a* in $H_2O_2$-treated mice was lower than that of control mice. The mean raw reads count of *Cldn8* in SB43 mice was higher than that in $H_2O_2$-treated mice. No obvious differences were observed for proteoglycan core proteins and tight junction proteins. There were no previous reports on IC/PBS for GAG-synthesizing enzymes, but differences were found for *Hyal1* and *Xylt2*.

A prestigious work by Gonzalez et al, reported that SB505124 improved urinary dysfunction by inhibiting hyperexcitation of bladder afferent nerves in cyclophosphamide-cystitis model [5,6]. In the present study, SB431542 was found to have the best therapeutic effect among TGF-β inhibitors for the cystitis, despite being at the same concentration as other TGF-β inhibitors. SB431542 and SB505124 inhibited ALK4, 5, and 7, respectively, while Repsox had specific inhibition of ALK5, therefore inhibition of ALK5 may contribute to urinary function. However, we assume that the greater efficacy of SB431542 than Repsox and SB505124 is due to differences in the degree of inhibition of ALK4,7 and CK1α, β, δ [42–44].

TGF-β inhibitors have potential applications in a wide range of diseases. Preclinical and clinical trials have shown their usefulness in treatment for fibrosis and tumors, especially by enlarging to existing cancer therapies such as radiation, chemotherapy and tumor vaccines. However, TGF-β inhibitors has been reported to have tumorigenic potential, such as reversible cutaneous keratoacanthoma/squamous cell carcinoma and hyperkeratosis [45]. In the present study, the blood samples showed no adverse effects on liver and kidney functions (ALT level is significantly elevated in $H_2O_2$/SB431542-treated group compared to control and the $H_2O_2$/saline-treated group, but the change is within the normal range) (S5 Fig). Also, no obvious adverse events such as weight loss were observed in the cystitis mice (S6 Fig). To avoid adverse effects of TGF-β inhibitors, it may be safe to pre-select the patient population based on surrogate markers of TGF-β involvement in the disease process and possible adverse contraindications before initiating therapy [7]. Peripheral blood may provide a useful non-invasive marker, such as the ability to quantitatively screen a patient's response to TGF-β inhibition based on measurement of P-SMAD2 levels in peripheral blood mononuclear cells [46]. In addition, when TGF-β inhibitors are actually administered to IC/BPS patients, it would be preferable to inject the TGF-β inhibitor locally into the bladder wall, like Botox injections for intractable overactive bladder. Alternatively, TGF-β inhibitor may be infused into the bladder, similar to intravesical infusion of DMSO to treat Hunner type interstitial cystitis, to avoid side effects of systemic administration, as a unique administration route specific to the bladder [47].

## Supporting information

**S1 Fig. Method of Voided Stain On Paper (VSOP) is shown.** The surface area is made by fluid drops of known volume (1 μL, 2.5μL, 5 μL, 10 μL, 20 μL, 50 μL, 100 μL, 150 μL, 200 μL, 250 μL, 300 μL, 350 μL, 400 μL, 500 μL). Linear correlation between liquid volume and stained area on the filter paper (r2 = 0.9924, y = 0.1401x - 20.766).
(DOCX)

**S2 Fig. mRNA expressions of TGF-β are elevated in $H_2O_2$-induced mice.** Effect of intravesical infusion of $H_2O_2$ on the mRNA expression levels of inflammatory cytokines. The mRNA expression levels of IL1β and TGF-β1,2 in control, saline, $H_2O_2$ groups (respectively n = 3) were measured by real-time RT-qPCR. Each mRNA level was normalized to the β-actin mRNA level and expressed relative to the control (fold). Results are represented as means ± sd. *P < 0.05, **P < 0.01.
(DOCX)

**S3 Fig. The stain sizes of the each group.** Scale bar = 50mm.
(DOCX)

**S4 Fig. Expression of mRNAs related to barrier function is shown.** The raw reads count of mRNAs related to barrier function in control, $H_2O_2$/saline-treated and $H_2O_2$/SB431542-treated groups (respectively n = 3) were measured by RNA-seq analyses. Results are represented as means ± sd. No statistically significant differences are shown because raw reads

count is used.
(DOCX)

**S5 Fig. Biological toxicity in liver and kidney is not confirmed.** Blood samples were collected for the assessment of liver and renal toxicity. sham, $H_2O_2$: intravesical $H_2O_2$ injection+ intraperitoneal saline injection, SB431542: intravesical $H_2O_2$ injection+ intraperitoneal SB431542 injection, respectively n = 4. Results are represented as means ± sd. *$P<0.05$. AST, aspartate aminotransferase; ALT, alanine aminotransferase; BUN, blood urea nitrogen; Cre, creatinine.
(DOCX)

**S6 Fig. Body weight and bladder weight.** Bladder weight (mg) on post-instillation day 10. sham, $H_2O_2$: intravesical $H_2O_2$ injection+ intraperitoneal saline injection, inhibitors(Repsox, SB431542, and SB505124): intravesical $H_2O_2$ injection+ intraperitoneal inhibitors injection, respectively n = 4. Chronological changes in whole body weight(g) on baseline, $H_2O_2$ pre-instillation day 0 and post-instillation day 4, 7. Results are represented as means ± sd. N.S.: no significant difference.
(DOCX)

## Acknowledgments

We thank Ms. Emi Homura for proofreading.

## Author Contributions

**Conceptualization:** Hideto Taga, Tsunao Kishida, Yuta Inoue, Osamu Ukimura, Osam Mazda.

**Data curation:** Hideto Taga, Kenta Yamamoto, Shin-ichiro Kotani.

**Formal analysis:** Hideto Taga, Kenta Yamamoto, Shin-ichiro Kotani.

**Investigation:** Hideto Taga, Shin-ichiro Kotani, Tsujimoto Masashi.

**Methodology:** Hideto Taga, Tsunao Kishida.

**Project administration:** Hideto Taga.

**Supervision:** Tsunao Kishida, Yuta Inoue, Osam Mazda.

**Validation:** Hideto Taga.

**Visualization:** Hideto Taga.

**Writing – original draft:** Hideto Taga.

**Writing – review & editing:** Osamu Ukimura, Osam Mazda.

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
