## [Decision Letter · Decision Letter 0]

11 Aug 2023

PONE-D-23-23604TGF-β inhibitor treatment of H₂O₂-induced cystitis models provides biochemical mechanism for elucidating interstitial cystitis/painful bladder syndrome patients.PLOS ONE

Dear Dr. Inoue,

Thank you for submitting your manuscript to PLOS ONE. After careful consideration, we feel that it has merit but does not fully meet PLOS ONE’s publication criteria as it currently stands. Therefore, we invite you to submit a revised version of the manuscript that addresses the points raised during the review process.

We look forward to receiving your revised manuscript.

Kind regards,

Yung-Hsiang Chen, Ph.D.

Academic Editor

PLOS ONE

Journal Requirements:

Additional Editor Comments:

Thank you for submitting the following manuscript to PLOS ONE.

Please revise the manuscript according to the reviewers' comments and upload the revised file.

Reviewers' comments:

Reviewer's Responses to Questions

**Comments to the Author**

1. Is the manuscript technically sound, and do the data support the conclusions?

Reviewer #1: No

Reviewer #2: Yes

2. Has the statistical analysis been performed appropriately and rigorously? 

Reviewer #1: Yes

Reviewer #2: Yes

3. Have the authors made all data underlying the findings in their manuscript fully available?

Reviewer #1: Yes

Reviewer #2: Yes

4. Is the manuscript presented in an intelligible fashion and written in standard English?

Reviewer #1: Yes

Reviewer #2: Yes

5. Review Comments to the Author

Reviewer #1: This is a well done and comprehensive study. The needed changes are relatively minor. The strengths are several and the weaknesses are mostly in the data and manuscript presentation. Strengths include the model itself. Other models of bladder inflammation including protamine, dilute HCl, and cyclophosphamide are as mentioned resolved within a week. The analysis of micturation patterns and gene expression add strength. Changes that need to be made are as follows. The material at the top of page 12 needs to be in the materials and methods. Figure S3 would be very helpful in the manuscript rather than in supporting material. However, a wealth of information on gene expression should be presented. These include genes involved with bladder barrier function such as uroplakins, proteoglycan core proteins identified previously as being involved in IC/BPS patients, tight junction proteins, claudins and GAG-synthesizing enzymes.

Reviewer #2: Comments on PONE-D-23-23604

Title: TGF-β inhibitor treatment of H₂O₂-induced cystitis models provides biochemical mechanism for elucidating interstitial cystitis/painful bladder syndrome patients.

The study rationale was sound.

The study design was well designed.

The experiments were well conducted. The results were well explained.

Some concerns existed on Fig. 4 H2 O2 infusion augmented expression of Trpa1, IL-1β, Col7a1, Krt13, and Krt6a genes in the bladder, which was reversed by SB431542 administration. Although the genes mentioned above were reversed by TGF-beta inhibitors, whether these were the mediators of H2 O2-induced cystitis remains elusive. The reasons need to be explained with caution. Similar situations happened in Fig. 5

6. PLOS authors have the option to publish the peer review history of their article (what does this mean?). If published, this will include your full peer review and any attached files.

Reviewer #1: No

Reviewer #2: No

---

## [Author Response · Author response to Decision Letter 0]

7 Oct 2023

September, 2023

PLOS ONE, Academic Editor Yung-Hsiang Chen, Ph.D.

Manuscript No.: PONE-D-23-23604

Title: TGF-β inhibitor treatment of H₂O₂-induced cystitis models provides biochemical mechanism for elucidating interstitial cystitis/painful bladder syndrome patients.

Dear Prof. Yung-Hsiang Chen

Thank you for reviewing our above-referenced manuscript. We appreciate the valuable comments from the editorials and reviewers. We sincerely appreciate this opportunity to address the reviewers’ constructive criticisms and to revise our manuscript accordingly. The specific comments of the reviewers have been addressed in “Response to Reviewers” below.

Certainly, if there are any other questions, please do not hesitate to contact us.

Best regards

Hideto Taga, M.D. and Osam Mazda, M.D., PhD.

"Response to Reviewers ".

●Reviewer #1 (Remarks to the Author): 

Question1

This is a well done and comprehensive study. The needed changes are relatively minor. The strengths are several and the weaknesses are mostly in the data and manuscript presentation. Strengths include the model itself. Other models of bladder inflammation including protamine, dilute HCl, and cyclophosphamide are as mentioned resolved within a week. The analysis of micturation patterns and gene expression add strength. 

Answer to the Question1

We appreciate the reviewer’s comment. 

Question2

Changes that need to be made are as follows. The material at the top of page 12 needs to be in the materials and methods. 

Answer to the Question2

We have taken into account the reviewer's feedback and opted to exclude the sentence in Results and added comments in Materials and Methods:

Results (page 12, lines 6-9)

We first examined a H₂O₂-induced cystitis model that we prepared based on previous reports with slight modifications.[3] Urinary frequency and one-time voiding volume were measured on days 1 and 8 to evaluate the effect of the H₂O₂ irritation on urinary function.

Materials and Methods (page 7, lines 9-11):

Urinary frequency and one-time voiding volume were measured on days 1 and 8 to evaluate the effect of the H₂O₂ irritation on urinary function.

Question3

Figure S3 would be very helpful in the manuscript rather than in supporting material.

Answer to the Question3

According to the reviewer #1’s comment, we have rephrased Figure S3 to Figure 2 in the Results section. Therefore, the figure numbers have been repositioned. 

Question4

 However, a wealth of information on gene expression should be presented. These include genes involved with bladder barrier function such as uroplakins, proteoglycan core proteins identified previously as being involved in IC/BPS patients, tight junction proteins, claudins and GAG-synthesizing enzymes.

Answer to the Question4

We have made bar graph with control, H₂O₂/saline-treated and H₂O₂/SB431542-treated groups by RNA-seq analyses in Supporting information (page 35, lines10-14) and added comments in Discussion (page 26-27) and References (page 26-27):

Supporting information (page 35, lines10-14)

S4 Fig. Expression of mRNAs related to barrier function is shown.The raw reads count of mRNAs related to barrier function in control, H₂O₂/saline-treated and H₂O₂/SB431542-treated groups (respectively n=3) were measured by RNA-seq analyses. Results are represented as means ± sd. No statistically significant differences are shown because raw reads count is used.

Discussion (page 26, line18 page 27, lines1-12)

According to previous reports, proteins involved in urothelial barrier function include uroplakins, proteoglycan core proteins, tight junction proteins, and claudins.[1, 41] Uroplakin knockout mice have been used as mouse models of IC/PBS.[41] S4 Fig shows that the mean raw reads count of H₂O₂-treated mice was lower than that of control mice and SB43 mice, except for Upk1b. The mean raw reads count of UPK3a in H₂O₂-treated mice was lower than that of control mice. The mean raw reads count of Cldn8 in SB43 mice was higher than that in H₂O₂-treated mice. No obvious differences were observed for proteoglycan core proteins and tight junction proteins. There were no previous reports on IC/PBS for GAG-synthesizing enzymes, but differences were found for Hyal1 and Xylt2. The detail mechanism by which TGF-β inhibitors protect the urothelial cell damage is still unknown.

References (page 33, lines20-25)

41. Akiyama Y, Luo Y, Hanno PM, Maeda D, Homma Y. Interstitial cystitis/bladder pain syndrome: The evolving landscape, animal models and future perspectives. Int J Urol. 2020; 27(6): 491-503. https://doi.org/10.1111/iju.14229. PMID: 32246572.

●Reviewer #2 (Remarks to the Author): 

Question1

The study rationale was sound. The study design was well designed. The experiments were well conducted. The results were well explained.

Answer to the Question1

We appreciate the reviewer’s comment. 

Question2

Some concerns existed on Fig. 4 H2O2 infusion augmented expression of Trpa1, IL-1β, Col7a1, Krt13, and Krt6a genes in the bladder, which was reversed by SB431542 administration. Although the genes mentioned above were reversed by TGF-beta inhibitors, whether these were the mediators of H2O2-induced cystitis remains elusive. The reasons need to be explained with caution.

Answer to the Question2

We understand the reviewer’s concern. We have added comments in Discussion and References.

Discussion (page 22, lines16-18)

Oyama et al. reported that mRNA expression of Trpa1 in bladders of the H₂O₂-treated mice was significantly higher than that in the control mice.[17]

Discussion (page 24, lines3-5)

Minami et al. reported that mRNA expression of IL-1β in bladders of the H₂O₂-treated mice was significantly higher than that in the control mice.[29]

Discussion (page 25, lines10-14 )

Although Col7a1 has not been reported as a mediator of H₂O₂-induced cystitis, the results of increased Col7a1 expression in H₂O₂-induced cystitis mice are consistent because the bladder wall of H₂O₂-induced cystitis mice has been shown to have increased fibrosis.[29]

Discussion (page 26, lines11-18)

Krt6 and Krt13 has not been reported as a mediator of H₂O₂-induced cystitis. The Krt6 expression is seen in stratified epithelial cells featuring hyperproliferation, such as psoriasis. Krt13 is associated with epithelial cell differentiation and signal transduction. [40] H₂O₂ destroys urothelial cells, followed by regeneration of urothelial cells through a process of proliferation and differentiation. Therefore, it seems that these increases in mRNA expression of keratins in H₂O₂-induced cystitis mice are consistent.

References (page 31,32,33)

17. Oyama S, Dogishi K, Kodera M, Kakae M, Nagayasu K, Shirakawa H, et al. Pathophysiological Role of Transient Receptor Potential Ankyrin 1 in a Mouse Long-Lasting Cystitis Model Induced by an Intravesical Injection of Hydrogen Peroxide. Front Physiol. 2017; 8: 877. https://doi.org/10.3389/fphys.2017.00877. PMID: 29249972.

29. Minami A, Tanaka T, Otoshi T, Kuratsukuri K, Nakatani T. Hyperbaric oxygen significantly improves frequent urination, hyperalgesia, and tissue damage in a mouse long-lasting cystitis model induced by an intravesical instillation of hydrogen peroxide. Neurourol Urodyn. 2019; 38(1): 97-106. https://doi.org/10.1002/nau.23822. PMID: 30411813

40. Dmello C, Srivastava SS, Tiwari R, Chaudhari PR, Sawant S, Vaidya MM. Multifaceted role of keratins in epithelial cell differentiation and transformation. J Biosci. 2019; 44(2). PMID: 31180046. 

Question3

Similar situations happened in Fig. 5

Answer to the Question3

We understand the reviewer’s concern. We have added comments in Discussion.

Discussion (page 24, lines17,18 page 25, line1)

Minami et al. showed that fibrosis area in bladder submucosa of the H₂O₂-treated mice increased than that of the control mice. [29]

---

## [Decision Letter · Decision Letter 1]

24 Oct 2023

TGF-β inhibitor treatment of H₂O₂-induced cystitis models provides biochemical mechanism for elucidating interstitial cystitis/painful bladder syndrome patients.

PONE-D-23-23604R1

Dear Dr. Inoue,

We’re pleased to inform you that your manuscript has been judged scientifically suitable for publication and will be formally accepted for publication once it meets all outstanding technical requirements.

Kind regards,

Yung-Hsiang Chen, Ph.D.

Academic Editor

PLOS ONE

Additional Editor Comments (optional):

Congratulations on the acceptance of your manuscript, and thank you for your interest in submitting your work to PLOS ONE.

Reviewers' comments:

Reviewer's Responses to Questions

**Comments to the Author**

1. If the authors have adequately addressed your comments raised in a previous round of review and you feel that this manuscript is now acceptable for publication, you may indicate that here to bypass the “Comments to the Author” section, enter your conflict of interest statement in the “Confidential to Editor” section, and submit your "Accept" recommendation.

Reviewer #1: All comments have been addressed

Reviewer #3: All comments have been addressed

2. Is the manuscript technically sound, and do the data support the conclusions?

Reviewer #1: Yes

Reviewer #3: Yes

3. Has the statistical analysis been performed appropriately and rigorously? 

Reviewer #1: Yes

Reviewer #3: Yes

4. Have the authors made all data underlying the findings in their manuscript fully available?

Reviewer #1: Yes

Reviewer #3: Yes

5. Is the manuscript presented in an intelligible fashion and written in standard English?

Reviewer #1: Yes

Reviewer #3: Yes

6. Review Comments to the Author

Reviewer #1: (No Response)

Reviewer #3: The paper reports a study on the effects of three TGF-β inhibitors, Repsox, SB431542, and SB505124, on a mouse model of interstitial cystitis/painful bladder syndrome (IC/PBS) induced by hydrogen peroxide (H₂O₂). The paper shows that TGF-β inhibitor treatment improved urinary function and histological changes in the IC/PBS mouse model, and SB431542 was the most effective among the TGF-β inhibitors. The paper also performs RNA-seq analysis to reveal the molecular mechanisms underlying the therapeutic effect of SB431542, and suggests that it involves the modulation of nociceptive mechanisms, immunity and inflammation, fibrosis, and urothelial function. The authors concluded that TGF-β inhibitors may be a promising therapy for IC/PBS, and that multiple pathways are involved in the improvement of urinary function by TGF-β inhibitor. The revision of the manuscript, no additional comments.

7. PLOS authors have the option to publish the peer review history of their article (what does this mean?). If published, this will include your full peer review and any attached files.

Reviewer #1: No

Reviewer #3: **Yes: **Wen-Chi Chen

---

## [Editor Report · Acceptance letter]

27 Oct 2023

PONE-D-23-23604R1 

TGF-β inhibitor treatment of H₂O₂-induced cystitis models provides biochemical mechanism for elucidating interstitial cystitis/painful bladder syndrome patients. 

Dear Dr. Inoue:

I'm pleased to inform you that your manuscript has been deemed suitable for publication in PLOS ONE. Congratulations! Your manuscript is now with our production department. 

Kind regards, 

on behalf of

Dr. Yung-Hsiang Chen 

Academic Editor

PLOS ONE